# Month of birth and risk of autism spectrum disorder: a retrospective cohort of male children born in Israel

Hila Shalev,[1] Ido Solt,[1,2] Gabriel Chodick[3,4]

► Prepublication history and additional material are available online. To view these files please visit the journal online (http://dx.doi.org/10.1136/bmjopen-2016-014606).

[1]Faculty of Medicine, Technion, Haifa, Israel
[2]Rambam Health Care campus, Haifa, Israel
[3]Maccabi Healthcare Services, Tel Aviv, Israel
[4]Sackler Faculty of Medicine, Tel Aviv University, Tel Aviv, Israel

**Correspondence to**
Prof. Gabriel Chodick;
hodik_g@mac.org.il

## ABSTRACT

**Background** Increased incidence and prevalence of autism spectrum disorder (ASD) over the last two decades have prompted considerable efforts to investigate its aetiological factors. We examined an association between month of birth and ASD incidence.

**Methods** In a retrospective cohort of male children born from January 1999 to December 2008 in a large health organisation in Israel (Maccabi Healthcare Services), ASD was followed from birth through December 2015.

**Results** Of 108 548 boys, 975 cases of ASD were identified. The highest rates (10.3 and 10.2 per 1000 male live births) were recorded for children born in May and August, respectively, and the lowest rates for February (7.6 per 1000 male live births). Among lower socioeconomic status households, boys born in August were more likely (OR=1.71; 95% CI 1.06 to 2.74) of being diagnosed with ASD than children born in January. Significantly higher rates were not observed for other months.

**Conclusions** In line with several previous studies, we found a modestly higher likelihood of autism occurrence among male children of lower socioeconomic levels born in August.

## INTRODUCTION

Autism spectrum disorder (ASD) is a biologically based neurodevelopmental condition characterised by (A) deficits in social communication and social interaction, and (B) restricted repetitive patterns of behaviour, interests and activities.[1] The prevalence of ASD has been rising[2] globally, and in Israel in particular.[3]

Analyses of season of birth or conception can provide clues about causes of diseases with unknown aetiology as in ASD. Patterns of seasonal variation in births have been found in several disorders; schizophrenia, mood disorders, epilepsy, language disorders, attention deficit hyperactivity disorder and neurodevelopmental disabilities.[4–9] An association between season of birth or conception suggests periodicity of an environmental aetiological factor active during the prenatal, perinatal or early postnatal phase of development.[10] The seasonal effects observed in

### Strength and limitations of this study

► Validated source of information of autism spectrum disorder (ASD) diagnosis.
► The use of systematic data that were collected prospectively on a large population.
► No information was available regarding ASD disease severity.
► nly boys were included, contrasting with other studies that included both sexes.
► Lack of information in our population on influenza infections and vaccines.

these disorders led researchers to examine whether the same would apply to ASD. The findings of such investigations are inconsistent; several studies claimed increased risk of ASD among individuals born or conceived in various seasons[11–13] or specific months,[14–21] while other reports found no seasonal patterns.[14 22–24] Inconsistencies in the literature could be due, in part, to a wide range of study populations, geographical areas, case definitions, comparison groups and analytical approaches.

In one of the first studies on season of birth and ASD to be carried out in a region other than the USA and western Europe, a study in Israel[18] found an increase in March and August births among 188 ASD cases when compared with the general population. The replication of seasonal patterns in a region with a climate and environment different from those of the countries previously studied supports the generalisability of the findings. However, a later examination of all ASD cases (n=211) among Jews in Israel over 5 years revealed no seasonal patterns in birth period. Such inconsistency can be explained by methodological differences, selection of comparison group, lack of statistical power and use of different diagnostic criteria.

The objective of the present study was to examine a relationship between autism and seasonality in a large population in Israel,

using the computerised databases of a single health provider and its standardised coding system. Male to female ratio among patients with ASD is reported between 3 to 1 and 4 to 1,[25–27] reflecting a disorder of potential differences in mechanisms and aetiology between the two sexes.[28–31] We excluded the girls from this investigation as we were interested in controlling the impact of sex by restricting the sample to boys.

## METHODS
### Settings
The 1994 National Health Insurance Law mandates compulsory universal health coverage for all Israeli residents in a health maintenance organisation (HMO) of their choice. Israeli citizens are obliged to choose one of four non-for-profit HMO's in the Israeli medical system, who provide equivalent medical services that are based on national medical standards. For this review, we examined data on ASD diagnosis from the computer database of the second-largest HMO, Maccabi Healthcare Services (MHS), which provides care to 2 million people (31% under 18 years of age) (Israeli Central Bureau of Statistics).

### Study population
Using the computerised databases of MHS, we identified a total of 147 171 male children who were born between January 1999 and December 2008 (approximately 20% of all 736 thousand live male births in Israel during the same years). Children's parents were identified using data on household members in MHS. Children with missing paternal data (mostly due to previously belonging to a different health provider) were excluded from the study. Comparison of excluded (n=38 875) and eligible (n=110 093) children showed no statistically significant (p>0.05) differences in mean birth weight (3271 g and 3315 g, respectively), mean gestational week (39.0 weeks and 39.1 weeks) or mean mother's age at birth (29.4 years and 30.5 years) and a similar distribution of socioeconomic status (SES). To avoid inaccurate paternal identification, an additional 1545 children were excluded because more than one man was registered in the child's household. Hence, the total study population was 108 548 boys of 83 452 households.

### Case identification and matching
A case was defined as a child with a physician-recorded diagnosis of autistic disorder, Asperger's syndrome orpervasive developmental-not otherwise specified (PDD-NOS), based on the Diagnostic and Statistical Manual of Mental Disorders, Fifth Edition (American Psychiatric Association 1994). In MHS, children with suspected ASD are referred to child development centres for a multidisciplinary assessment for diagnosis validation. The MHS registry of patients with ASD and validation processes were previously described.[32]

### Other study variables
SES was assessed according to the poverty index of the child's address enumeration area, as defined by the 2008 national census.[33] The Poverty Index is based on several parameters including household income, educational qualifications, crowding, material conditions, and car ownership, census of population and housing. SES levels range between 1 (lowest) to 10 (highest). In this study SES below median (6) was considered as low SES. Calendar year and month of conception was estimated based on the child's date of birth and the length of gestation as reported in the birth certificate. We categorised season of birth (and conception) as winter comprising December, January and February; spring, March, April and May; summer, June, July and August; and fall, September, October and November. Month of conception was calculated by subtracting gestational week from the date of birth.

### Statistical analyses
First, bivariate analyses were conducted between month of birth and other covariates previously reported to be associated with increased risk of autism.[34 35] The covariates included gestational age in weeks as a continuous variable,[36] birth weight, older maternal or paternal age (>35 years), and SES level above median (>6). Gestational age was available for 97 264 individuals (90% of the study population). Missing gestational ages were imputed using multiple imputation (five imputations) with birth weight as a covariate. The Cochran-Armitage test was used to assess a significant trend in the association between month of birth and ASD.

Multivariable logistic regression analyses were performed to determine the OR and 95% CI for ASD by month of birth, using January as the reference month. For the purpose of comparison with previous studies, we fitted alternative logistic regression models, with season of birth, month and season of conception (rather than birth), first month of the second trimester(see online supplementary appendix table a) and third trimester (see online supplementary appendix table b) as the main independent variables. Additionally, to evaluate the potential effect modification by socioeconomic level, the study population was divided into two groups in further analyses: low–medium (1–6) and medium–high (7–10). Effect modification by month of birth and SES was investigated by including a cross-product interaction term between the exposure variable (birth in August × SES group). A p value for interaction[37] was computed to determine the statistical significance of adjusted ORs related to birth in August between high and low SES levels using WINPEPI statistical package.[38]

In a sensitivity analysis, using birth month and season, we excluded one calendar year at a time to identify the effect of one specific year, with no significant results.

A two-tailed p value <0.05 was considered statistically significant. All data processing and statistical analyses were performed using IBM SPSS Statistics for Windows, V.22.0. (IBM Corp. Armonk, New York, USA).

## RESULTS
The mean number of monthly births ranged from 8318 in February to 9654 in August. In bivariate analyses, number

of births, maternal age, paternal age, birth weight, gestational length and SES were not associated with the month of birth (table 1). Of the 108 548 boys born during the study period, 975 were diagnosed with ASD, representing an incidence rate of 9.0 per 1000 live births. Mean age at ASD diagnosis was 3.0 years (SE=0.07 years). The highest incidences of ASD were recorded among children born in May and August: 10.3 and 10.2 per 1000, respectively, while the lowest incidences were in January and February (7.6 and 8.1 per 1000, respectively) (table 2). The results by month of conception are shown in table 3.

In a multivariable model, birth months of August and May were associated with a non-significantly (p=0.16 and p=0.14, respectively) increased OR (table 4). When the data were stratified by household socioeconomic level, the birth month of August in low SES was associated with an OR of 1.71 (95% CI 1.06 to 2.74) of ASD. No other significant results were observed for other birth months or in children of higher SES (table 5 and see online supplementary tables a,b,c,d,e) . There was a borderline effect modification between birth in August and SES level (p=0.07). Using similar multivariable analyses, we found no significant (p>0.5) association between ASD and season of birth (see online supplementary appendix table c), season of conception (see online supplementary appendix table d), or month of conception (supplementary file 1).

## DISCUSSION

In this retrospective cohort of boys born in Israel during a period spanning 10 years, children of low SES together with birth in August were associated with an increased likelihood of being diagnosed with ASD. This is the one of the largest studies outside the USA to describe the relationship between ASD incidence and month of birth.

The literature with respect to season of birth and risk of ASD is inconsistent. Of the 15 studies (total of 31 777 births) summarised in table 6 (all conducted in the northern hemisphere) 5 studies (total n=23 355) reported an increased risk of ASD among individuals born in August. March was significantly associated with increased risk in eight studies (total n=3008), and three studies (total n=5839) found no significant association between month of birth and ASD occurrence. No other specific birth month was repeatedly reported to be associated with ASD.

There are several mechanisms by which season of birth may affect the developing brain, either directly on the fetus and newborn or indirectly through the mother during pregnancy or the postpartum period. Seasonal environmental factors that could be associated with ASD include viral or other infections,[39 40] nutritional factors[41] and vitamin deficiencies.[42] The outcome of exposure to prenatal viral infection depends on many factors: the variation in susceptibility of the maternal and fetal host, including the maternal immune status, the infecting virus, the strain of virus, the developmental stage of the

**Table 1** Study population characteristics by month of birth

| Maternal age | | | | Paternal age | | | | Birth weight, g | | | | Gestational age | | | | SES 1 to 10 | | | | Births | Month |
|---|---|---|---|---|---|---|---|---|---|---|---|---|---|---|---|---|---|---|---|---|---|
| M (SD) | Med | 25% | 75% | M (SD) | Med | 25% | 75% | M (SD) | Med | 25% | 75% | M (SD) | Med | 25% | 75% | M (SD) | Med | 25% | 75% | | |
| 30.1 (4.9) | 30 | 27 | 33 | 32.9 (5.7) | 33 | 29 | 36 | 3296 (525) | 3320 | 3000 | 3630 | 39.1 (1.9) | 39 | 38 | 40 | 6.1 (2.3) | 6 | 4 | 8 | 9367 | January |
| 30.2 (4.9) | 30 | 27 | 34 | 32.9 (5.7) | 33 | 29 | 36 | 3296 (528) | 3320 | 3000 | 3640 | 39.0 (2.0) | 39 | 38 | 40 | 6.2 (2.3) | 6 | 5 | 8 | 8318 | February |
| 30.2 (4.9) | 30 | 27 | 34 | 32.9 (5.6) | 33 | 29 | 36 | 3305 (519) | 3340 | 3020 | 3650 | 39.0 (1.9) | 39 | 38 | 40 | 6.1 (2.3) | 6 | 4 | 8 | 9038 | March |
| 30.3 (4.9) | 30 | 27 | 34 | 33.0 (5.6) | 33 | 29 | 37 | 3310 (519) | 3340 | 3030 | 3640 | 39.1 (1.9) | 39 | 38 | 40 | 6.2 (2.3) | 6 | 5 | 8 | 8465 | April |
| 30.3 (4.9) | 30 | 27 | 34 | 33.0 (5.6) | 33 | 29 | 37 | 3300 (532) | 3330 | 3010 | 3640 | 39.1 (1.9) | 39 | 38 | 40 | 6.3 (2.3) | 6 | 5 | 8 | 8711 | May |
| 30.3 (5.0) | 30 | 27 | 34 | 33.1 (5.6) | 33 | 29 | 37 | 3306 (538) | 3350 | 3010 | 3650 | 39.0 (2.0) | 39 | 38 | 40 | 6.2 (2.3) | 6 | 5 | 8 | 8523 | June |
| 30.2 (4.9) | 30 | 27 | 34 | 33.0 (5.7) | 33 | 29 | 37 | 3304 (528) | 3330 | 3010 | 3650 | 39.0 (2.0) | 39 | 38 | 40 | 6.1 (2.3) | 6 | 4 | 8 | 9330 | July |
| 30.3 (4.9) | 30 | 27 | 34 | 33.1 (5.7) | 33 | 29 | 37 | 3312 (532) | 3340 | 3020 | 3660 | 39.0 (1.9) | 39 | 38 | 40 | 6.1 (2.3) | 6 | 4 | 8 | 9654 | August |
| 30.3 (4.9) | 30 | 27 | 34 | 33.1 (5.7) | 33 | 29 | 37 | 3291 (537) | 3320 | 3000 | 3640 | 39.0 (1.9) | 39 | 38 | 40 | 6.1 (2.3) | 6 | 4 | 8 | 9270 | September |
| 30.3 (5.0) | 30 | 27 | 34 | 33.0 (5.7) | 33 | 29 | 37 | 3302 (524) | 3330 | 3020 | 3640 | 39.1 (1.9) | 39 | 38 | 40 | 6.1 (2.3) | 6 | 4 | 8 | 9521 | October |
| 30.3 (5.0) | 30 | 27 | 34 | 33.1 (5.7) | 33 | 29 | 37 | 3302 (532) | 3330 | 3000 | 3640 | 39.0 (1.9) | 39 | 38 | 40 | 6.1 (2.3) | 6 | 4 | 8 | 9069 | November |
| 30.3 (5.0) | 30 | 27 | 34 | 33.0 (5.7) | 33 | 29 | 37 | 3288 (540) | 3320 | 3000 | 3630 | 39.0 (2.0) | 39 | 38 | 40 | 6.1 (2.3) | 6 | 4 | 8 | 9282 | December |
| **30.3 (5)** | **30** | **27** | **33.9** | **36.8** | **33** | **29** | **33.0 (5.7)** | **3642.5** | **3330.8** | **3010** | **3301 (529.5)** | **39.0 (2.0)** | **39** | **38** | **40** | **6.1 (2.3)** | **6** | **4.33** | **8** | **9046** | **Mean** |

SES, socioeconomic status.

**Table 2**  Incidence rate and 95% confidence interval (CI) per 1000 male live births of autism spectrum disorder (ASD), by month of birth

| Month of birth | ASD cases | Rate (per 1000) | 95% CI* | |
|---|---|---|---|---|
| January | 76 | 8.1 | 6.44 | 10.09 |
| February | 63 | 7.6 | 5.87 | 9.62 |
| March | 74 | 8.2 | 6.48 | 10.21 |
| April | 74 | 8.7 | 6.92 | 10.90 |
| May | 90 | 10.3 | 8.36 | 12.62 |
| June | 78 | 9.2 | 7.29 | 11.35 |
| July | 89 | 9.5 | 7.71 | 11.67 |
| August | 98 | 10.2 | 8.29 | 12.30 |
| September | 80 | 8.6 | 6.89 | 10.67 |
| October | 85 | 8.9 | 7.18 | 10.97 |
| November | 84 | 9.3 | 7.44 | 11.40 |
| December | 84 | 9.0 | 7.27 | 11.14 |
| Total | 975 | 8.96 | 7.17 | 11.07 |

*See text for details.

fetus, the amount of virus reaching the fetus—particularly the central nervous system and immune system, the route of access, genetics and probably other factors.[43] Maternal viral infection has been cited as the 'principal non-genetic cause of autism'.[44] Maternal infections (ie, rubella, cytomegalovirus, influenza), prolonged fever and maternal inflammation cause changes in a variety of inflammatory cytokines that have been associated with ASD.[45 46] Depending on the location of immune activation, persistent viral infection could lead to chronically elevated cytokine levels that can be produced directly in the brain or gain access to the CNS by crossing an immature blood-brain barrier, altering CNS development.[47]

A few studies found influenza infection during pregnancy to be a factor that increases risk for infantile autism[39 48] especially in the first trimester.[46] Results from animal models also support an effect of maternal influenza infection on the neurodevelopment of the offspring.[49 50] In this analysis, ASD was related significantly with month of birth but not with month or season of conception and therefore our results do not support this potential aetiology. This is in line with recently published results of a large cohort from the USA.[51] Potential explanations can be proposed for the observed effect modification of increased risk of ASD among August born of lower SES. These may include higher likelihood

**Table 3**  Incidence rate and 95% confidence interval (CI) per 1000 male live births of autism spectrum disorder (ASD), by month of conception

| Month of conception | ASD cases | Rate (per 1000) | 95% CI* | |
|---|---|---|---|---|
| January | 87 | 9.2 | 7.4 | 11.3 |
| February | 74 | 8.7 | 6.9 | 11 |
| March | 77 | 8.3 | 6.6 | 10.4 |
| April | 72 | 8.0 | 6.3 | 10.1 |
| May | 77 | 8.4 | 6.7 | 10.5 |
| June | 68 | 7.8 | 6.1 | 9.9 |
| July | 72 | 8.2 | 6.4 | 10.3 |
| August | 89 | 10.2 | 8.2 | 12.5 |
| September | 91 | 10.9 | 8.8 | 13.4 |
| October | 79 | 8.4 | 6.6 | 10.4 |
| November | 99 | 10.8 | 8.8 | 13.1 |
| December | 87 | 9.1 | 7.3 | 11.2 |
| Total | 972 | 9.0 | 8.4 | 9.9 |

*See text for details.

**Table 4** Crude and adjusted odds ratio (OR) and 95% confidence interval (CI) of autism spectrum disorder (ASD) for month of birth: multivariable logistic regression

| Month of birth | OR crude | 95% CI | | OR adjusted* | 95% CI | |
|---|---|---|---|---|---|---|
| January | 1 (ref.) | | | 1 (ref.) | | |
| February | 0.93 | 0.67 | 1.31 | .92 | .66 | 1.29 |
| March | 1.01 | 0.73 | 1.39 | 1.00 | 0.73 | 1.38 |
| April | 1.08 | 0.78 | 1.49 | 1.08 | 0.78 | 1.48 |
| May | 1.28 | 0.94 | 1.73 | 1.26 | 0.92 | 1.71 |
| June | 1.13 | 0.82 | 1.55 | 1.11 | 0.81 | 1.52 |
| July | 1.16 | 0.86 | 1.58 | 1.15 | 0.84 | 1.56 |
| August | 1.25 | 0.93 | 1.69 | 1.24 | 0.92 | 1.68 |
| September | 1.07 | 0.78 | 1.46 | 1.05 | 0.77 | 1.44 |
| October | 1.10 | 0.81 | 1.50 | 1.09 | 0.80 | 1.49 |
| November | 1.13 | 0.83 | 1.55 | 1.12 | 0.82 | 1.53 |
| December | 1.11 | 0.81 | 1.51 | 1.09 | 0.80 | 1.50 |

*Adjusted for child's age, father's age, mother's age, district, birth weight, gestational age with imputation (see text for details).

of maternal exposure to infections or particulate matter air pollution that have been connected to increased risk of ASD.[52] A role of vitamin D deficiency in the development of autism has been proposed.[42] Thus, seasonal fluctuations in vitamin D should also be considered. In the northern hemisphere, vitamin D levels are the lowest in winter time;[53–55] however, a study of healthy volunteers from Israel found no differences in vitamin D deficiency rates between summer and winter.[56]

Strengths of the current study include the investigation of a large and unselected population that is similar in its characteristics (mean maternal age, years of follow-up, etc) to a large cohort of a recently published US study that examined autism and seasonality,[51] the cohort design,

the use of systematic data that were collected prospectively, validated source of information of ASD diagnosis, the availability of data on prematurity and the objective assessment of SES. Some limitations should be discussed. We had no information about the ASD disease severity; in one previous report,[57] high functioning children with ASD showed an increase in summer births compared with the low functioning children with ASD. In addition, SES assessment was conducted using data on current residential area, which is not necessarily the residential area during the time of pregnancy, nor necessarily identical to individual SES. The children who were born at the end of the year were younger than those born earlier in the year, but this is negligible because they were all at

**Table 5** Crude and adjusted odds ratios (ORs) and 95% confidence interval (CI) from multivariable logistic regression of autism spectrum disorder (ASD) for month of birth, by socioeconomic level (SES)

| Month of birth | SES low | | | | | | SES high | | | | | |
|---|---|---|---|---|---|---|---|---|---|---|---|---|
| | OR crude | 95% CI | | OR adjusted* | 95% CI | | OR crude | 95% CI | | OR adjusted* | 95% CI | |
| January | 1 (ref.) | | | 1 (ref.) | | | 1 (ref.) | | | 1 (ref.) | | |
| February | 1.01 | 0.58 | 1.75 | 1.01 | 0.58 | 1.75 | 0.89 | 0.58 | 1.35 | 0.88 | 0.57 | 1.34 |
| March | 1.34 | 0.80 | 2.22 | 1.34 | 0.81 | 2.22 | 0.82 | 0.54 | 1.25 | 0.81 | 0.53 | 1.24 |
| April | 1.30 | 0.77 | 2.19 | 1.32 | 0.78 | 2.22 | 0.93 | 0.62 | 1.40 | 0.93 | 0.62 | 1.40 |
| May | 1.31 | 0.78 | 2.20 | 1.30 | 0.78 | 2.19 | 1.19 | 0.81 | 1.74 | 1.17 | 0.80 | 1.72 |
| June | 1.26 | 0.75 | 2.12 | 1.25 | 0.74 | 2.11 | 1.01 | 0.67 | 1.51 | 0.99 | 0.66 | 1.49 |
| July | 1.37 | 0.83 | 2.25 | 1.36 | 0.82 | 2.25 | 1.01 | 0.68 | 1.49 | 0.99 | 0.67 | 1.48 |
| August | 1.70 | 1.06 | 2.73 | 1.71 | 1.07 | 2.76 | 1.00 | 0.67 | 1.48 | 0.99 | 0.66 | 1.46 |
| September | 1.32 | 0.80 | 2.19 | 1.32 | 0.80 | 2.18 | 0.92 | 0.61 | 1.38 | 0.91 | 0.60 | 1.36 |
| October | 1.24 | 0.75 | 2.05 | 1.24 | 0.75 | 2.07 | 1.03 | 0.69 | 1.52 | 1.01 | 0.68 | 1.50 |
| November | 1.45 | 0.88 | 2.37 | 1.46 | 0.89 | 2.40 | 0.96 | 0.64 | 1.44 | 0.93 | 0.62 | 1.40 |
| December | 1.49 | 0.91 | 2.43 | 1.51 | 0.92 | 2.46 | 0.89 | 0.59 | 1.34 | 0.88 | 0.58 | 1.32 |

*Adjusted for child's age, father's age, mother's age, district, birth weight, age with imputation (see text for details).

**Table 6** Summary of findings in previous studies investigating the relationship between seasonal variations of birth and autism

| Ref. | Study period | Place | ASD cases | Months of increased risk of ASD |
|------|-------------|-------|-----------|--------------------------------|
| 11 | 1990–2002 | California | 19 238 | November, December, August, September, October |
| 14 | 1992–2000 | California | 2940 | August* |
| 14 | 1997–1999 | California | 3897 | No association |
| 22 | 1947–1980 | UK | 1631 | No association |
| 15 | 1983–2002 | Maryland | 1068 | Spring |
| 16 | 1981*** | North Carolina | 810 | March, August |
| 23 | 1947–1992 | Child dev. clinic | 620 | No association** |
| 17 | 1945–1980 | Denmark | 328 | March |
| 24 | 1980–1985 | Israel | 211 | No association |
| 18 | 1964–1986 | Israel | 188 | March and August |
| 12 | 1980s | Ontario , Canada | 179 | Summer and spring |
| 19 | 1977–1985 | USA | 175 | March |
| 20 | 1962–1984 | Sweden | 174 | March |
| 13 | 1972–1978 | Japan | 132 | Spring–summer (April, May, June) |
| 21 | 1990–1992 | UK | 86 | Spring |

*Conception in November (for years 1994–1996).
**(p<0.1 for middle of August for verbal ASD).
***Year of publication.
ASD, autism spectrum disorder.

the age of at least 6 years when the study was done and most autistic children are diagnosed by this age.[58] Our analysis included only boys and therefore generalisability to both sexes is limited. Data were unavailable regarding important risk factors such as family history, viral infections and environmental factors. Further investigations on whether the observed seasonal patterns interplay with genetic and environmental risk factors in the development of ASD may provide clues regarding the aetiology of these complex disorders.

## CONCLUSIONS

In line with several previous studies, we found modestly higher likelihood of ASD occurrence among male children of lower SES levels born in August. Month of birth may indicate seasonal causal factors that could be associated with autism. The fact that these patterns appear to differ by SES may also be aetiologically significant.

**Correction notice** This article has been corrected since it was published online. Data sharing statement has been updated.

**Acknowledgements** This study is part of the MD thesis by HS.

**Contributors** HS and GC contributed substantially to the conception of the work, analysis, interpretation of data for the work, drafting the work and final approval of the version to be published. IS contributed substantially to the acquisition, interpretation of data for the work, revising it critically for important intellectual content and final approval of the version to be published.

**Competing interests** None declared.

**Provenance and peer review** Not commissioned; externally peer reviewed.

**Data sharing statement** No Additional data are available.

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

**APPENDIX**

**Table A** Crude and adjusted odd ratio (OR) and 95% confidence interval from multivariable logistic regression of autistic spectrum disorder (ASD) for first month of second trimester, by socioeconomic status (SES)

| Month of second trimester | OR crude | 95% CI | | OR adjusted* | 95% CI | | OR crude | 95% CI | | OR adjusted* | 95% CI | |
|---|---|---|---|---|---|---|---|---|---|---|---|---|
| January | 1 (ref.) | | | 1 (ref.) | | | 1 (ref.) | | | 1 (ref.) | | |
| February | 0.83 | 0.51 | 1.35 | 0.83 | 0.51 | 1.36 | 1.03 | 0.67 | 1.57 | 1.03 | 0.67 | 1.57 |
| March | 1.08 | 0.69 | 1.69 | 1.08 | 0.69 | 1.69 | 0.87 | 0.57 | 1.35 | 0.86 | 0.56 | 1.33 |
| April | 0.89 | 0.55 | 1.43 | 0.89 | 0.55 | 1.43 | 1.12 | 0.74 | 1.69 | 1.11 | 0.73 | 1.67 |
| May | 0.68 | 0.41 | 1.12 | 0.68 | 0.41 | 1.13 | 0.99 | 0.65 | 1.51 | 0.98 | 0.64 | 1.49 |
| June | 0.87 | 0.53 | 1.41 | 0.87 | 0.54 | 1.41 | 1.09 | 0.72 | 1.64 | 1.07 | 0.71 | 1.63 |
| July | 1.00 | 0.62 | 1.59 | 0.98 | 0.61 | 1.56 | 1.02 | 0.67 | 1.56 | 1.04 | 0.68 | 1.59 |
| August | 0.74 | 0.44 | 1.23 | 0.72 | 0.43 | 1.20 | 0.97 | 0.63 | 1.49 | 0.97 | 0.63 | 1.49 |
| September | 0.94 | 0.58 | 1.54 | 0.93 | 0.57 | 1.51 | 0.86 | 0.55 | 1.35 | 0.87 | 0.55 | 1.36 |
| October | 0.84 | 0.51 | 1.37 | 0.83 | 0.51 | 1.37 | 1.10 | 0.73 | 1.66 | 1.11 | 0.73 | 1.67 |
| November | 0.84 | 0.51 | 1.40 | 0.83 | 0.50 | 1.38 | 1.25 | 0.84 | 1.87 | 1.24 | 0.83 | 1.86 |
| December | 0.83 | 0.51 | 1.36 | 0.83 | 0.51 | 1.36 | 1.10 | 0.73 | 1.65 | 1.09 | 0.72 | 1.64 |

*Adjusted for child's age, father's age, mother's age, district, birth weight, age with imputation (see text for details).

**Table B** Crude and adjusted odd ratio (OR) and 95% confidence interval from multivariable logistic regression of autistic spectrum disorder (ASD) for first month of third trimester, by socioeconomic status (SES)

| Month of second trimester | OR crude | 95% CI | | OR adjusted* | 95% CI | | OR crude | 95% CI | | OR adjusted* | 95% CI | |
|---|---|---|---|---|---|---|---|---|---|---|---|---|
| January | 1 (ref.) | | | 1 (ref.) | | | 1 (ref.) | | | 1 (ref.) | | |
| February | 1.04 | 0.61 | 1.77 | 1.04 | 0.61 | 1.78 | 1.22 | 0.81 | 1.82 | 1.21 | 0.81 | 1.81 |
| March | 0.97 | 0.57 | 1.63 | 0.97 | 0.58 | 1.64 | 0.96 | 0.63 | 1.46 | 0.95 | 0.63 | 1.45 |
| April | 1.19 | 0.73 | 1.95 | 1.20 | 0.74 | 1.97 | 1.13 | 0.75 | 1.68 | 1.12 | 0.75 | 1.68 |
| May | 0.86 | 0.51 | 1.46 | 0.87 | 0.52 | 1.48 | 0.86 | 0.56 | 1.32 | 0.86 | 0.56 | 1.31 |
| June | 1.26 | 0.77 | 2.04 | 1.27 | 0.78 | 2.06 | 0.84 | 0.55 | 1.30 | 0.83 | 0.54 | 1.27 |
| July | 0.99 | 0.60 | 1.65 | 1.01 | 0.61 | 1.67 | 1.07 | 0.71 | 1.61 | 1.05 | 0.70 | 1.58 |
| August | 0.78 | 0.45 | 1.33 | 0.79 | 0.46 | 1.36 | 0.99 | 0.66 | 1.51 | 0.98 | 0.64 | 1.48 |
| September | 1.12 | 0.68 | 1.86 | 1.12 | 0.68 | 1.86 | 1.12 | 0.75 | 1.69 | 1.12 | 0.74 | 1.69 |
| October | 0.84 | 0.49 | 1.44 | 0.83 | 0.48 | 1.42 | 0.83 | 0.53 | 1.29 | 0.83 | 0.54 | 1.30 |
| November | 0.92 | 0.53 | 1.57 | 0.91 | 0.53 | 1.56 | 1.00 | 0.65 | 1.53 | 1.00 | 0.65 | 1.54 |
| December | 1.07 | 0.64 | 1.80 | 1.08 | 0.64 | 1.81 | 0.81 | 0.52 | 1.26 | 0.81 | 0.52 | 1.26 |

*Adjusted for child's age, father's age, mother's age, district, birth weight, age with imputation (see text for details).

**Table C** Crude and adjusted odd ratio (OR) and 95% confidence interval from multivariable logistic regression of autistic spectrum disorder (ASD) for season of birth, by socioeconomic status (SES)

| Season of birth | SES median-high | | | | | | SES median-low | | | | | |
|---|---|---|---|---|---|---|---|---|---|---|---|---|
| | OR crude | 95% CI | | OR adjusted | 95% CI | | OR crude | 95% CI | | OR adjusted | 95% CI | |
| Winter | 1 (ref.) | | | 1 (ref.) | | | 1 (ref.) | | | 1 (ref.) | 1 (ref.) | |
| Spring | 1.12 | 0.84 | 1.50 | 1.12 | 0.84 | 1.50 | 1.06 | 0.83 | 1.34 | 1.06 | 0.83 | 1.34 |
| Summer | 1.24 | 0.94 | 1.63 | 1.23 | 0.93 | 1.63 | 1.08 | 0.85 | 1.37 | 1.08 | 0.85 | 1.37 |
| Fall | 1.14 | 0.86 | 1.51 | 1.14 | 0.86 | 1.51 | 1.05 | 0.82 | 1.33 | 1.04 | 0.82 | 1.32 |

*Adjusted for child's age, father's age, mother's age, district, birth weight, age with imputation.

**Table D** Crude and adjusted odd ratio (OR) and 95% confidence interval from multivariable logistic regression of autistic spectrum disorder (ASD) for season of conception, by socioeconomic status (SES)

| Season of conception | SES median-high | | | | | | SES median-low | | | | | |
|---|---|---|---|---|---|---|---|---|---|---|---|---|
| | OR crude | 95% CI | | OR adjusted* | 95% CI | | OR crude | 95% CI | | OR adjusted* | 95% CI | |
| Winter | 1 (ref.) | | | 1 (ref.) | | | 1 (ref.) | | | 1 (ref.) | | |
| Spring | 1.00 | 0.75 | 1.34 | 0.99 | 0.74 | 1.33 | 0.96 | 0.75 | 1.22 | 0.96 | 0.75 | 1.23 |
| Summer | 1.00 | 0.74 | 1.35 | 0.99 | 0.73 | 1.33 | 0.96 | 0.75 | 1.22 | 0.97 | 0.76 | 1.24 |
| Fall | 1.09 | 0.82 | 1.45 | 1.08 | 0.82 | 1.44 | 1.05 | 0.83 | 1.33 | 1.06 | 0.83 | 1.34 |

*Adjusted for child's age, father's age, mother's age, district, birth weight, age with imputation.

**Table E** Crude and adjusted odd ratio (OR) and 95% confidence interval from multivariable logistic regression of autistic spectrum disorder (ASD) for month of conception, by socioeconomic status (SES)

| Month of conception | SES 1-6 | | | | | | SES 7-10 | | | | | |
|---|---|---|---|---|---|---|---|---|---|---|---|---|
| | OR crude | 95% CI | | OR adjusted* | 95% CI | | OR crude | 95% CI | | OR adjusted* | 95% CI | |
| January | 1 (ref.) | | | 1 (ref.) | | | 1 (ref.) | | | 1 (ref.) | | |
| February | 0.89 | 0.52 | 1.50 | 0.89 | 0.52 | 1.51 | 1.09 | 0.72 | 1.65 | 1.08 | 0.72 | 1.63 |
| March | 1.13 | 0.70 | 1.85 | 1.14 | 0.70 | 1.86 | 0.84 | 0.55 | 1.30 | 0.84 | 0.54 | 1.29 |
| April | 0.90 | 0.53 | 1.53 | 0.89 | 0.53 | 1.52 | 1.06 | 0.71 | 1.60 | 1.09 | 0.72 | 1.64 |
| May | 1.05 | 0.63 | 1.75 | 1.03 | 0.62 | 1.71 | 0.94 | 0.62 | 1.44 | 0.95 | 0.62 | 1.45 |
| June | 1.19 | 0.72 | 1.96 | 1.17 | 0.71 | 1.93 | 0.74 | 0.46 | 1.16 | 0.75 | 0.47 | 1.18 |
| July | 0.79 | 0.45 | 1.38 | 0.79 | 0.45 | 1.38 | 0.96 | 0.63 | 1.45 | 0.97 | 0.64 | 1.47 |
| August | 1.11 | 0.67 | 1.83 | 1.09 | 0.66 | 1.81 | 1.13 | 0.76 | 1.68 | 1.13 | 0.76 | 1.68 |
| September | 1.29 | 0.79 | 2.10 | 1.28 | 0.78 | 2.08 | 1.09 | 0.73 | 1.64 | 1.10 | 0.73 | 1.65 |
| October | 0.95 | 0.57 | 1.58 | 0.96 | 0.58 | 1.59 | 0.88 | 0.58 | 1.34 | 0.89 | 0.58 | 1.35 |
| November | 1.14 | 0.70 | 1.86 | 1.14 | 0.70 | 1.86 | 1.16 | 0.78 | 1.72 | 1.16 | 0.78 | 1.72 |
| December | 1.18 | 0.74 | 1.91 | 1.18 | 0.74 | 1.91 | 0.90 | 0.59 | 1.36 | 0.89 | 0.59 | 1.36 |

*Adjusted for child's age, father's age, mother's age, district, birth weight, age with imputation.see text for details

