## [Reviewer comments · BMJ Open]

ARTICLE DETAILS

TITLE (PROVISIONAL)	Month of birth and risk of autism spectrum disorder, a retrospective cohort of male children born in Israel
AUTHORS	Shalev, Hila; Solt, Ido; Chodick, Gabriel

VERSION 1 - REVIEW

REVIEWER	Igor Burstyn Drexel University, USA
REVIEW RETURNED	26-Oct-2016

GENERAL COMMENTS	The title MUST be edited to indicate that only males were studied. The study cannot claim to use unselected population – it only studied males! Please revise. The rationale for limiting studies to males due to excess of cases among males makes no sense, unless it is simply an issue of statistical power. Please finesse this point to avoid perception of gender bias, i.e. it sounds like you are saying that autism among girls is not important. Why is lack of data on flu and other vaccine mentioned as limitation? To be sure, some key variables were not measured but focus on vaccines in general makes no sense and produces a misleading impression of relevance of vaccination to ASD risk. You discuss this issue in a balanced way but the summary statements need to be more narrowly focused on your specific arguments. P5: Is there any information on SES indicators (listed top of P6) of excluded and included subjects? Dichotomizing gestational age at 37 weeks is not optimal for autism research (e.g. see doi: 10.1016/j.jpeds.2012.07.040, PMID: 22947654): please examine a range of gestational ages as confounders. Did you control for parental history of psychiatric disorders? This should be possible using health-providers' claim data. Please show results by month of conception in detail. You mention these analyses but do not give figures. Month of conception, after control for duration of gestation (e.g. as continuous variable) may well be a better proxy of events in etiologically relevant (early) gestation. You may also wish to check for effect modification by duration of gestation as it would affect to which months a pregnancy was exposed. Following up on suggestions above, I think that you may wish to do something less traditional and yet potentially more informative: study
---

exposure to specific month/season of each trimester of pregnancy. This may help illuminate whether, e.g. events associated with summer place one at greater risk for ASD depending on whether pregnancy is in 1st or 3rd trimester, etc.

I discourage uncritical citation of OR on environmental exposures bottom page 8, top page 9. There is simply no need to reach here for something as fantastical as interaction of months of birth with asbestos/styrene exposure in ASD, etc. There are many correlates of low SES and there is no need to overly focus on the ones you site, given that you have no evidence that in your study these specific stressors explain seasonality. Less is more here...

Why is history of vaccination not relevant to seasonality? You contradict yourself here, it seems. Fly vaccination is seasonal in all places where I have ever lived, so if it is different in Israel, this must be clarified.

I suggest that you drop adjective "significant" from concluding statements (and elsewhere as much as possible): whether p-value was less than 5% is the least interesting feature of your results (esp. in epidemiology where measures of association are of interest, not hypothesis tests) and the worldly SIGNIFICANCE of your observations is highly UNCERTAIN – it may prove very important in the future but we do not know this with any confidence.

There is something off with column width in table 1; please remedy.

Table 1: please show medians and inter-quartile ranges together with mean/SD – the distributions may not be normal and therefore not fully described by mean/SD.

Table 2: it would be great to have the same type of table by month of conception; it would help readability if denominators of rates (births) were also shown here, not just in Table 1.

Table 3: please show only 2 figures after decimal point on estimate of OR – there is simply no info in the third decimal and readability will be greatly improved.

Please create a figure like Tables 1 & 2 that shows high and low SES separately – reader needs to see key features of the data, not just summary of odds ratios as in Figure 1. You can keep the figure as it tells a nice story but it must be supported with tables.

Figure 1 may be misleading with respect to heterogeneity of effect by SES – please conduct more rigorous test of effect modification/heterogeneity; e.g. see [Kaufman JS, MacLehose RF. Which of these things is not like the others? *Cancer*. 2013 Dec 15;119(24):4216-22. doi: 10.1002/cncr.28359. PubMed PMID: 24022386; PubMed Central PMCID: PMC4026206] for guidance. Depending on what your more formal exploration of heterogeneity may reveal, you may wish to re-think your conclusions: perhaps evidence of effect modification by SES is not as strong as you initially thought or perhaps is stronger...

Figure 1: Add 95%CI for high SES estimates.

Figure 1: It seems wrong to connect point estimates by a line – you are not measuring looking for month-to-month (or week-to-week)

	trend of a continuous scale. It is my view that it is best to represent monthly estimates by points; this will also allow (though use of offsets) to show 95%CI for both SES groups on one graph.
--	---

REVIEWER	Keely Cheslack-Postava Columbia University Medical Center, USA
REVIEW RETURNED	26-Oct-2016

GENERAL COMMENTS	This manuscript describes a study examining the association between month of birth and ASD among males only using data from one of four HMOs in Israel, over 10 years of births. As described in the discussion of the paper, this topic has been examined in numerous prior studies; however, not in this population. The authors report a significantly elevated OR for ASD among August versus January births among those living in lower SES areas only, though it is not clear whether this estimate differs significantly from that for higher SES areas (see below). The paper could be improved by giving a specific hypothesis. This would better guide the analysis and strengthen the conclusions to be drawn. For example, seasonality is mentioned, but only month is used as a measure of exposure, whereas "season" might incorporate multiple months, but the means of aggregating those months would depend on what was the relevant 'seasonal' exposure (sunlight, temperature, infections, etc.) Also, the timing of birth and the timing of a relevant exposure during gestation will not be the same, so discussion of what might be the relevant "time window" during gestation should be included. Additional concerns and suggestions are given below.  1. Abstract – Results: there must be an error as the rate described as higher 1/1000 is lower than that described as lower (7.6/1000) 2. Introduction, p. 4, last paragraph. ASD should not be described as a disorder of sexual differentiation. 3. Potential differences in etiology in males vs. females would be a reasonable rationale for conducting separate/stratified analyses, but would not require excluding females. This should be better rationalized. 4. Can people switch HMO membership, and if so, will diagnoses be missed for those who switched after birth? 5. The authors should formally test and present the results for whether the associations between ASD and birth month differed by SES (i.e. test for interaction). They could also consider additional analyses (tertiles of SES, use a different measure of SES) to assess the robustness of the purported association. 6. Various possible mechanisms for explaining association of birth/gestational timing with ASD are given in the discussion – the paper would be strengthened if these could better be used to contextualize the current study – which of these explanations are/are not consistent with the results that the authors observed and why? For example, with respect to pollutants, seasonal patterns of pollutant levels were not discussed. 7. The analysis assumes the relationship (ASD and birth month) is constant over 10 years of the study, which is not necessarily a given (see Mazumdar et al, 2012)- this should be stated or discussed.
---

VERSION 1 – AUTHOR RESPONSE

Reviewer: 1

1. The title MUST be edited to indicate that only males were studied

- Corrected

2. The study cannot claim to use unselected population – it only studied males! Please revise. The rationale for limiting studies to males due to excess of cases among males makes no sense, unless it is simply an issue of statistical power. Please finesse this point to avoid perception of gender bias, i.e. it sounds like you are saying that autism among girls is not important.

- In MHS, the male-to-female ratio among ASD patients is 5:1 (Journal of Autism and Developmental Disorders, 2013;43:785–793). The expected number of approximately 200 ASD female cases would yield low statistical power (less than 20%) for detecting a minimal odds ratio of 1.25 at 0.05 significance. Hence, the analysis was restricted to males. We added an explanation in the text.

3. Why is lack of data on flu and other vaccine mentioned as limitation? To be sure, some key variables were not measured but focus on vaccines in general makes no sense and produces a misleading impression of relevance of vaccination to ASD risk. You discuss this issue in a balanced way but the summary statements need to be more narrowly focused on your specific arguments.

- Since we do not have information on vaccination in our database, we were unable to address this interesting question. We deleted the text regarding speculations on the role of vaccination.

4. P5: Is there any information on SES indicators (listed top of P6) of excluded and included subjects?

- We found no meaningful differences in the distribution of SES between included and excluded patients (please see below)

-

1st SES quartile

Included : 26.2%

Excluded: 22.4%

2nd SES quartile

Included : 24.6%

Excluded: 26.7%

3rd SES quartile

Included : 27.3%

Excluded: 29.0%

4th SES quartile

Included : 21.9%

Excluded: 22.0%

5. Dichotomizing gestational age at 37 weeks is not optimal for autism research (e.g. see doi: 10.1016/j.jpeds.2012.07.040, PMID: 22947654): please examine a range of gestational ages as confounders.

- Following the reviewer's important comment, we used the gestational week covariate as a continuous variable with no predefined cutoffs.

6. Did you control for parental history of psychiatric disorders? This should be possible using health-providers' claim data.

- Paternal psychiatric disorders would have been of great value; however, we were not granted an IRB approval to collect such sensitive data.

7. Please show results by month of conception in detail. You mention these analyses but do not give figures. Month of conception, after control for duration of gestation (e.g. as continuous variable) may well be a better proxy of events in etiologically relevant (early) gestation. You may also wish to check for effect modification by duration of gestation as it would affect to which months a pregnancy was exposed.

- We examined several seasonal models including month of conception (defined as the calendar month of the date of birth minus the gestational week), season of conception, and season of birth. None of these models indicated a significant association. Crude and adjusted odds ratios from multivariable models stratified by SES level are now given in appendix tables a, b, c.

8. Following up on suggestions above, I think that you may wish to do something less traditional and yet potentially more informative: study exposure to specific month/season of each trimester of pregnancy. This may help illuminate whether, e.g. events associated with summer place one at greater risk for ASD depending on whether pregnancy is in 1st or 3rd trimester, etc.

- Following the reviewer's suggestion, we examined whether the first months of the second and third trimester are associated with an increased ASD risk. Results by SES level are given in tables d and e in the Appendix. None of these models yielded significant results.

9. I discourage uncritical citation of OR on environmental exposures bottom page 8, top page

9. There is simply no need to reach here for something as fantastical as interaction of months of birth with asbestos/styrene exposure in ASD, etc. There are many correlates of low SES and there is no need to overly focus on the ones you site, given that you have no evidence that in your study these specific stressors explain seasonality. Less is more here...

- The paragraph was deleted. Thank you.

10. Why is history of vaccination not relevant to seasonality? You contradict yourself here, it seems. Flu vaccination is seasonal in all places where I have ever lived, so if it is different in Israel, this must be clarified.

- Since we do not have information on vaccination in our database, we were unable to address this interesting question. We deleted the text regarding speculations on the role of vaccination.

11. I suggest that you drop adjective "significant" from concluding statements (and elsewhere as much as possible): whether p-value was less than 5% is the least interesting feature of your results (esp. in epidemiology where measures of association are of interest, not hypothesis tests) and the worldly SIGNIFICANCE of your observations is highly UNCERTAIN – it may prove very important in the future but we do not know this with any confidence.

- Corrected

12. There is something off with column width in table 1; please remedy.

- Corrected

13. Table 1: please show medians and inter-quartile ranges together with mean/SD – the distributions may not be normal and therefore not fully described by mean/SD.

- Corrected

14. Table 2: it would be great to have the same type of table by month of conception; it would help readability if denominators of rates (births) were also shown here, not just in Table 1.

- Corrected

15. Table 3: please show only 2 figures after decimal point on estimate of OR – there is simply no info in the third decimal and readability will be greatly improved.

- Corrected

16. Please create a figure like Tables 1 & 2 that shows high and low SES separately – reader needs to see key features of the data, not just summary of odds ratios as in Figure 1. You can keep the figure as it tells a nice story but it must be supported with tables.

- Corrected. See new table in the file (table 5, page 18), figure 1 was erased since it contains the same information.

17. Figure 1 may be misleading with respect to heterogeneity of effect by SES – please conduct more rigorous test of effect modification/heterogeneity; e.g. see [Kaufman JS, MacLehose RF. Which of these things is not like the others? *Cancer*. 2013 Dec 15;119(24):4216-22. doi: 10.1002/cncr.28359. PubMed PMID: 24022386; PubMed Central PMCID: PMC4026206] for guidance. Depending on what your more formal exploration of heterogeneity may reveal, you may wish to re-think your conclusions: perhaps evidence of effect modification by SES is not as strong as you initially thought or perhaps is stronger...

- A formal test for effect modification between birth in August and SES was added to the revised version. See the new text in the Methods (Page 7 Lines 7-11) and Results (Page 8 Lines 6-7).

18. Figure 1: Add 95%CI for high SES estimates. Figure 1: It seems wrong to connect point estimates by a line – you are not measuring looking for month-to-month (or week-to-week) trend of a continuous scale. It is my view that it is best to represent monthly estimates by points; this will also allow (though use of offsets) to show 95%CI for both SES groups on one graph.

- Figure 1 was erased since it contains the same information as table 5 (page 18)

Reviewer: 2

1. The paper could be improved by giving a specific hypothesis. This would better guide the analysis and strengthen the conclusions to be drawn. For example, seasonality is mentioned, but only month is

used as a measure of exposure, whereas “season” might incorporate multiple months, but the means of aggregating those months would depend on what was the relevant ‘seasonal’ exposure (sunlight, temperature, infections, etc.) Also, the timing of birth and the timing of a relevant exposure during gestation will not be the same, so discussion of what might be the relevant “time window” during gestation should be included.

- A new set of analyses has been added, including results for month of conception, season of conception, season of birth, and the first calendar months of the second and third trimesters. None yielded significant results.

2. Additional concerns and suggestions are given below.

2.1 Abstract – Results: there must be an error as the rate described as higher 1/1000 is lower than that described as lower (7.6/1000)

- Please see correction on page 2 lines 8-9

2.2 Introduction, p. 4, last paragraph. ASD should not be described as a disorder of sexual differentiation. Potential differences in etiology in males vs. females would be a reasonable rationale for conducting separate/stratified analyses, but would not require excluding females. This should be better rationalized.

- In MHS, the male-to-female ratio among ASD patients is 5:1 (Journal of Autism and Developmental Disorders, 2013;43:785–793). With expected number of approximately 200 ASD female cases, the statistical power to detect a minimal odds ratio of 1.25 at 0.05 significance would be very low (less than 20%). Hence, the analysis was restricted to males.

3. Can people switch HMO membership, and if so, will diagnoses be missed for those who switched after birth?

- Children were required to have at least three years of consecutive membership in MHS. Annual retention rate in MHS is very high (~99%).

4. The authors should formally test and present the results for whether the associations between ASD and birth month differed by SES (i.e. test for interaction). They could also consider additional analyses (tertiles of SES, use a different measure of SES) to assess the robustness of the purported association.

- A formal test for effect modification between birth in August and SES is included in the revised manuscript. See the new text in the Methods (Page 7 Lines 7-11) and Results (Page 8 Lines 6-7).

5. Various possible mechanisms for explaining association of birth/gestational timing with ASD are given in the discussion – the paper would be strengthened if these could better be used to contextualize the current study – which of these explanations are/are not consistent with the results that the authors observed and why? For example, with respect to pollutants, seasonal patterns of pollutant levels were not discussed.

- The paragraph that mentioned pollutants was deleted.

6. The analysis assumes the relationship (ASD and birth month) is constant over 10 years of the study, which is not necessarily a given (see Mazumdar et al, 2012)- this should be stated or discussed.

- To examine the sensitivity of the regression model to patterns for individual years, we repeated the

analysis by excluding one calendar year at a time. Results remained similar in all analyses.

We look forward to hearing from you regarding our submission. We would be glad to respond to any further questions and comments that you may have.

VERSION 2 – REVIEW

REVIEWER	Igor Burstyn Drexel University, Philadelphia, PA, USA
REVIEW RETURNED	28-Nov-2016

GENERAL COMMENTS	Thank you for undertaking the revisions. I have two concerns. First, the argument about low power to study effects among girls is poorly motivated by the power calculation. This is in part because the power calculation is incomplete: it does not describe model assumed or the prevalence of exposure. But more fundamentally, test of hypothesis is not as important as evaluation of the magnitude of association in epidemiology. Thus, the question of power of a hypothesis test is of secondary importance. Lastly, 200 cases is plenty to test a question about etiology of autism given current state of the literature. Consequently, I would suggest to not make argument about power for excluding girls but maybe say that you were interested in controlling for effect of sex by restricting of the sample to boys. I strongly urge to follow-up with analysis if girls in the near future. My intuition is that such analysis is necessary on ethical grounds given your intriguing observation among boys. I appreciate additional statistical analyses and tests of effect modification. Please describe what you did in more detail, as it is unclear to me how the test of effect modification was constructed and how analyses for time-windows of exposure to "season" when pregnancy took place were conducted. There is not reason to economize on word count in the on-line publication, especially when it comes to essentials like methods. Please also try to purge word "significant" from the abstract as you did with statements of conclusions -- it will help convey your actual findings more accurately.
---

VERSION 2 – AUTHOR RESPONSE

Reviewer: 1

First, the argument about low power to study effects among girls is poorly motivated by the power calculation. This is in part because the power calculation is incomplete: it does not describe model assumed or the prevalence of exposure. But more fundamentally, test of hypothesis is not as important as evaluation of the magnitude of association in epidemiology. Thus, the question of power of a hypothesis test is of secondary importance. Lastly, 200 cases is plenty to test a question about etiology of autism given current state of the literature. Consequently, I would suggest to not make argument about power for excluding girls but maybe say that you were interested in controlling for effect of sex by restricting of the sample to boys. I strongly urge to follow-up with analysis if girls in the near future. My intuition is that such analysis is necessary on ethical grounds given your intriguing observation among boys.

Following your suggestion, we explained the restriction to males only as an interest in controlling the impact of sex on the results. Indeed, a few studies (e.g. Waterhouse et al., 2000) have suggested a differential effect of sex on seasonality. In our further investigations we will examine females specifically (page 5 lines 8-10)

I appreciate additional statistical analyses and tests of effect modification. Please describe what you did in more detail, as it is unclear to me how the test of effect modification was constructed and how analyses for time-windows of exposure to "season" when pregnancy took place were conducted. There is not reason to economize on word count in the on-line publication, especially when it comes to essentials like methods.

We have expanded explanations in the Methods and Results. The examined time windows included month and season of birth, month and season of conception, and second and third trimester. Only month of birth was significantly associated with ASD. (page 7 lines 6-7, lines 20-22 , page 8 lines 1-6, page 9 lines 3-4)

Please also try to purge word "significant" from the abstract as you did with statements of conclusions -- it will help convey your actual findings more accurately.

Thank you for your comment, we left out the word "significant".

Thank you for your comments, we look forward to hearing from you regarding our revised manuscript.

VERSION 3 – REVIEW

REVIEWER	Igor Burstyn Drexel University, USA
REVIEW RETURNED	17-Feb-2017

GENERAL COMMENTS	Thank you for undertaking further revisions. The paper is now a helpful contribution to understanding the ASD puzzle.
---

REVIEWER	Keely Cheslack-Postava Columbia University, United States
REVIEW RETURNED	08-Mar-2017

GENERAL COMMENTS	Introduction The expanded background added to the introduction is helpful, in particular with details about the previous studies conducted in Israel. However, some minor clarification is needed. p. 4, lines 34-37—the added sentence is not clear—if associations were found, but not to particular months, then to what? Also, perhaps “reported” would be a better term than “claimed” in this sentence. P. 4, line 44, delete “areas” p. 4, line 52, I suggest “generalizability” rather than “validity” given that the latter may refer to either internal or external validity. Methods
---

	p. 7, line 54- “the month of the second and first trimester” needs clarification as trimesters each span > 1 month. p. 9, sensitivity analysis – given the main finding is for month of birth, I would suggest conducting this analysis for birth month Results Table 1, please check the last row (“Total”). In particular, paternal age and gestational age, 75th percentiles are of the form of mean (SD) entries. Also, it’s not clear why some of the percentiles for the total row have decimals (is this row for the total population combined or were the monthly rows averaged?).
--	---

VERSION 3 – AUTHOR RESPONSE

- p. 4, lines 34-37—the added sentence is not clear—if associations were found, but not to particular months, then to what? Also, perhaps “reported” would be a better term than “claimed” in this sentence.

Thank you for your comment, the sentence was changed to:

“Several studies claimed increased risk of ASD among individuals born in various seasons or specific months of conception and birth, while other reports found no seasonal patterns” with compatible references.

- P. 4, line 44, delete “areas”

The word “areas” was deleted.

- p. 4, line 52, I suggest “generalizability” rather than “validity” given that the latter may refer to either internal or external validity.

The word “validity” has been replaced to the word “generalizability”.

METHODS

- p. 7, line 54- “the month of the second and first trimester” needs clarification as trimesters each span > 1 month.

Thank you for the comment, the sentence was changed to: “the first month of second and third trimester”.

- p. 9, sensitivity analysis – given the main finding is for month of birth, I would suggest conducting this analysis for birth month

Thank you for the comment, the sentence was wrong, the sensitivity analysis with calendar year was conducted only for the base model using birth month as the independent variable.

RESULTS

- Table 1, please check the last row (“Total”). In particular, paternal age and gestational age, 75th percentiles are of the form of mean (SD) entries. Also, it’s not clear why some of the percentiles for the total row have decimals (is this row for the total population combined or were the monthly rows averaged?).

Thank you for your comment. The table has been corrected. The last row provides population average.

We look forward to hearing from you regarding our submission. We would be glad to respond to any further questions and comments that you may have.

VERSION 4 – REVIEW

REVIEWER	Keely Cheslack-Postava Columbia University, United States
REVIEW RETURNED	29-Mar-2017

GENERAL COMMENTS	The comments have been adequately addressed and the paper is now acceptable for publication.
--